# Correlation-driven nonequilibrium exciton site transition in a WSe$_2$/WS$_2$ moiré supercell

Jinjae Kim [1,2,7], Jiwon Park[1,2,7], Hyojin Choi[1,2], Taeho Kim[3,4], Soonyoung Cha[5], Yewon Lee[3,4], Kenji Watanabe [6], Takashi Taniguchi [6], Jonghwan Kim [3,4], Moon-Ho Jo [3,4] & Hyunyong Choi [1,2] ✉

Moiré superlattices of transition metal dichalcogenides offer a unique platform to explore correlated exciton physics with optical spectroscopy. Whereas the spatially modulated potentials evoke that the exciton resonances are distinct depending on a site in a moiré supercell, there have been no clear demonstration how the moiré excitons trapped in different sites dynamically interact with the doped carriers; so far the exciton-electron dynamic interactions were presumed to be site-dependent. Thus, the transient emergence of nonequilibrium correlations are open questions, but existing studies are limited to steady-state optical measurements. Here we report experimental fingerprints of site-dependent exciton correlations under continuous-wave as well as ultrashort optical excitations. In near-zero angle-aligned WSe$_2$/WS$_2$ heterobilayers, we observe intriguing polarization switching and strongly enhanced Pauli blocking near the Mott insulating state, dictating the dominant correlation-driven effects. When the twist angle is near 60°, no such correlations are observed, suggesting the strong dependence of atomic registry in moiré supercell configuration. Our studies open the door to largely unexplored nonequilibrium correlations of excitons in moiré superlattices.

The intriguing phenomena of moiré superlattices emerge through the vertical alignment of two layers of two-dimensional (2D) materials, characterized by a relative twist angle or lattice mismatch. This structural arrangement introduces an additional periodic potential landscape, influencing the behavior of electrons and holes within a moiré supercell. Excitons and charged particles in the moiré supercell offer new avenues to explore profound correlation phenomena that include unconventional superconductivity, ferromagnetism, and the tunable manipulation of Wigner crystal and Mott insulating states[1–5]. Because of the strong light-matter interaction features of transition metal dichalcogenides (TMDs), a rich spectrum of excitonic entities has been unveiled in this context, with optical spectroscopy. Examples include the reflection contrast (RC), photoluminescence (PL), and helicity-dependent magnetic circular dichroism, whereby interesting optical characteristics encompassing moiré intralayer[4] and interlayer excitons[6], trions[7], moiré polarons[8], spin-polarons[9], and bosonic correlated excitons[10,11] have been studied.

Despite these advancements, investigations on the microscopic nonequilibrium dynamics of these excitonic species are still infancy. While existing studies are limited to steady-state optical measurements, the transient emergence of nonequilibrium correlations and the corresponding optical consequence remain unexplored. Examinations on such fields might provide new possibilities for the observation of correlation-driven nonequilibrium dynamics. Particularly, the investigation within the charge-transfer insulating states that possess the energetically favorable second local minimum potential in a moiré

[1]Department of Physics, Seoul National University, Seoul 08826, Korea. [2]Institute of Applied Physics, Seoul National University, Seoul 08826, Korea. [3]Department of Materials Science and Engineering, Pohang University of Science and Technology, Pohang 37673, Korea. [4]Center for van der Waals Quantum Solids, Institute for Basic Science (IBS), Pohang 37673, Korea. [5]Department of Physics and Astronomy, University of California, Riverside, CA 92521, USA. [6]Advanced Materials Laboratory, National Institute for Materials Science, 1-1 Namiki, Tsukuba 305-0044, Japan. [7]These authors contributed equally: Jinjae Kim, Jiwon Park. ✉e-mail: hy.choi@snu.ac.kr

supercell offers a compelling opportunity to unravel the impact of spatial atomic registry on the evolving spectral transients of excitons.

In this Article, we report the observation of moiré exciton site transitions driven by the strong correlation between doped electrons and photoexcited excitons. We performed RC, interlayer exciton PL, and time-resolved pump-probe spectroscopy in a gate-controlled, angle-aligned WSe₂/WS₂ heterobilayer. Our findings reveal that, upon n-doping, the polarization switching and the strongly enhanced Pauli blocking phenomena are consistent with the dynamic site transition of moiré excitons, which eventuates only in the R-stacked (near 0°) device (R1) (Fig. 1a). Conversely, the H-stacked (near 60°) device (H1) does not manifest such phenomena, indicating its strong dependence on a moiré supercell configuration. All measurements were conducted at a temperature of 4 K. We obtain consistent results from an additional R-stacked device R2 and R3. The main text of results is primarily from the R-stacked heterobilayer R1 and the H-stacked heterobilayer H1.

## Results

### Steady-state optical spectroscopy

In our experiment, we fabricated both R- and H-stacked WSe₂/WS₂ heterobilayers. A single-gated device is employed for both bilayers, as depicted in Fig. 1b. Notably, we opted for a gold bottom gate instead of the conventional graphene-based dual gate configuration. This strategic choice aimed to maximize the signal-to-noise ratio for our pump-probe spectroscopy while avoiding any possible nonequilibrium artifacts from graphene dynamics. In Fig. 1c, we presented an optical microscopy image of the device R1. The twist angle was confirmed

using the second harmonic generation (SHG) spectroscopy (Supplementary Fig. 1).

First, we examined the gate-dependent RC near the WSe₂ A exciton resonance (-1.7 eV) of the device R1 (Fig. 2a). Three distinct intralayer exciton peaks were observed at the gate voltage $V_G = 0V$, labelled as $X_1, X_2$, and $X_3$. With changing $V_G$, we have discerned a series of insulating states at integer fillings ($\nu$) that are consistent with many prior reports[4,12–14]. The same plots near the WS₂ A exciton resonance (-2.0 eV) are presented in Supplementary Fig. 2.

Given the type II band alignment characteristics of the WSe₂/WS₂ heterobilayer, we have proceeded to investigate the PL measurements of the interlayer excitons around the photon energy of 1.4 eV (Fig. 2b) under 632.8 nm continuous-wave laser excitation. Specifically, Fig. 2c shows the degree of circular polarization $\rho (= \frac{\sigma^+/\sigma^+ - \sigma^+/\sigma^-}{\sigma^+/\sigma^+ + \sigma^+/\sigma^-})$ derived from the polarization-resolved PL measurements. When $0<\nu<1$, the PL emission is co-circularly polarized, but makes a transition into the linearly-polarized PL when $\nu>1$. In contrast, the cross-circularly polarized PL features are seen across the entire negative $\nu$. The asymmetric PL peak shift, observed as a function of $V_G$, arises from the differences in the Stark effect[15]. These differences are specifically linked to our device structure, where the WS₂ layer is positioned closer to the back gate in our stacking configuration of heterobilayer (Supplementary Fig. 3). Due to the type-II alignment, when electrons are induced by the positive $V_G$, they are primarily populated in the WS₂ layer. This, in turn, screens the electric field between WSe₂ and WS₂ layers, resulting in a reduced Stark effect. To establish a comparative scope, we have conducted identical measurements on a device H1. Figure 2d and e shows

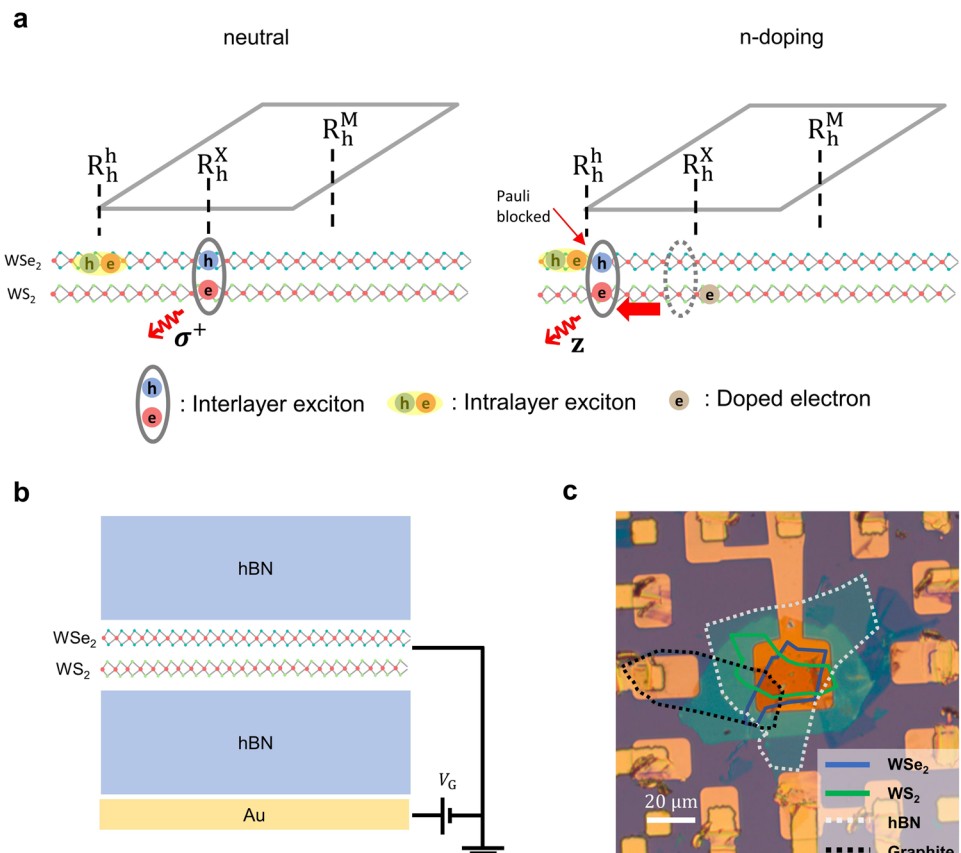

**Fig. 1 | Schematics of the moiré exciton site transition in WSe₂/WS₂ devices. a** Schematic shows the strong correlation induced moiré exciton site transition. At charge neutrality, the interlayer excitons occupy $R_h^X$ site where the optical selection rule for the interlayer exciton renders the co-circularly polarized PL emission (left). When doped with electrons, the Coulomb repulsion ($U_{e-ex}$) between electrons and interlayer excitons makes the interlayer excitons transfer to the $R_h^h$ site which is the secondary energy minimum of the moiré potential (right). Consequently, the polarization switching of the interlayer excitons occurs into the linearly-polarized PL. Correspondingly, the enhanced Pauli blocking is observed by both steady-state and ultrashort optical measurements. **b** A device schematic showing the hBN encapsulated WSe₂/WS₂ with a gold bottom-gate. **c** Optical microscopy image of a near-0° twisted WSe₂/WS₂ heterobilayer (device R1).

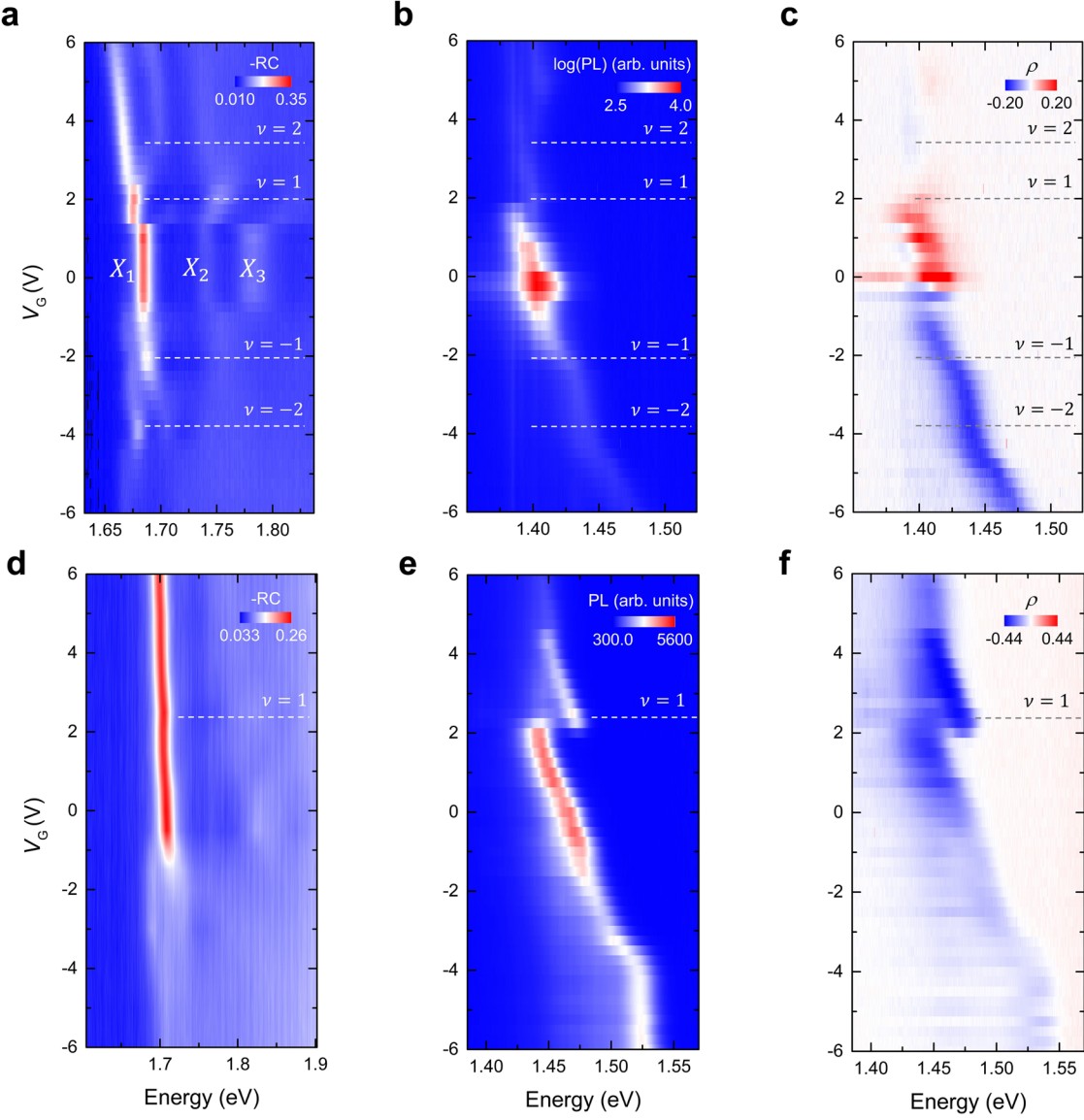

**Fig. 2 | Twist-angle-dependent degree of circular polarization of interlayer excitons. a** $V_G$-dependent reflection contrast (RC) spectrum of a R-stacked ($\theta \approx 1.2° \pm 0.6°$) WSe$_2$/WS$_2$ moiré superlattice (Device R1). Positive (negative) $V_G$ refers to electron (hole) doping. Filling factor $\nu$ refers to the number of doped carriers per moiré unit-cell (moiré density $n_0 = 2.19 \times 10^{12}$ cm$^{-2}$). **b** $V_G$-dependent interlayer exciton PL (log scale) of device R1 when the incident laser and detected PL are co-circularly polarized. **c** The corresponding degree of circular polarization ($\rho = \frac{\sigma^+/\sigma^+ - \sigma^+/\sigma^-}{\sigma^+/\sigma^+ + \sigma^+/\sigma^-}$) measured from the polarization resolved PL. Polarization switching phenomena are observed near Mott insulating states ($\nu = 1$) at $V_G = 2$V as well as $V_G = -0.24$V. **d–f** The same plots as **a–c** but for the H-stacked device H1. **f** Polarization switching is not observed but the cross-circularly polarized feature is seen for the entire $V_G$.

the $V_G$-dependent RC and PL, performed on the device H1. The PL peak shows a sudden blueshift, and the absorption shows a spectral kink at $\nu = 1$. As seen in Fig. 2f, $\rho$ does not show the polarization switching, but maintains the cross-polarized character.

Subsequently, we have investigated the laser intensity $I$ dependent $\rho$ of the interlayer PL emission on the device R1 (Fig. 3a). Here, we define the "threshold voltage", at which the circular polarization switching occurs – from the co-circularly polarized PL emission to the linearly-polarized (cross-circularly polarized) one. These threshold voltages are determined by the $V_G$-dependent valley polarization (Supplementary Fig. 4). Figure 3b summarizes the threshold voltage as a function of $I$. Here, we note that the threshold voltage for the n-doping case displays a relatively weak dependence on $I$, while it exhibits a much stronger $I$-dependence for the p-doping case; it represents that a higher $V_G$ is required for the polarization switching to occur for the p-doping case with increasing $I$.

Such distinct doping dependence of PL polarization arises from the contrasting atomic arrangements between the R- and the H-stacked heterobilayers. It has been well established that the absence of out-of-plane mirror symmetry leads to a unique optical selection rule for the interlayer electron-hole recombination[16–20]. This rule, governing not only the spin-valley indices but also the local stacking, plays a pivotal role in the PL polarization. The corresponding optical selection rules of interlayer transition are summarized in Supplementary Table 1 and Supplementary Table 2. In the case of the H-stacked device, where the interlayer electrons and holes experience lateral separation, the interlayer excitons experience two types of sites for the recombination to occur, namely H$_h^h$ or H$_h^X$. These sites possess opposite selection rules but maintain the spin preserving configuration. The observed cross-polarized PL suggests a higher recombination rate at the H$_h^h$ site by the optical selection rule[18]. Because the site transitions are not observed on the device H1, the emitted light helicity remains unchanged.

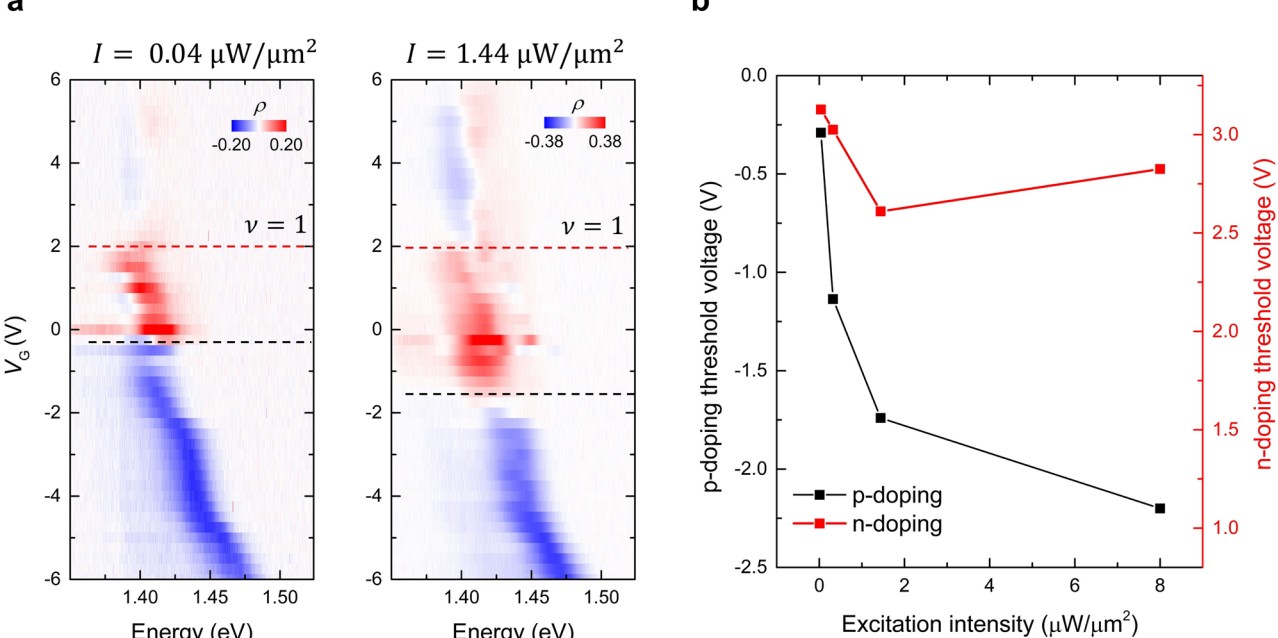

**Fig. 3 | Excitation intensity dependent threshold voltages. a** $V_G$-dependent degree of circular polarization ($\rho = \frac{\sigma^+/\sigma^+ - \sigma^+/\sigma^-}{\sigma^+/\sigma^+ + \sigma^+/\sigma^-}$) is shown for the two selected $I$. The red dashed lines indicate $\nu = 1$ and the black dashed lines indicate the threshold voltage when $V_G < 0$. **b** Threshold voltage as a function of $I$. The red and black lines indicate the threshold voltage for the polarization switching when $V_G > 0$ and $V_G < 0$, respectively. The polarization switching exhibits a strong dependence on $I$ when

$V_G < 0$; the higher $I$, the larger $V_G$ is required to switch the polarization. Such phenomena are due to the efficient valley-flip scattering as demonstrated in ref. [18]. When $V_G > 0$, however, we suggest a different mechanism, namely a correlation-driven exciton site transition. In this regime, the polarization switching shows a weak $I$-dependence. See the main text for more details.

For the R-stacked device, the recombination of the electron and hole takes place at the $R_h^X$ site, resulting in the co-circularly polarized PL. After $\sigma^+$ excitation, the photogenerated electrons in the K valley of $WSe_2$ undergo relaxation into both K and −K valleys of $WS_2$ through the spin-flip and the valley-flip scattering. A recent study[18] has elucidated such a feature in the p-doping regime; when the population of doped holes surpasses the imbalanced electron population via efficient valley-flip relaxation, polarization switching occurs from the co-circularly polarized PL to the cross-circularly polarized one. This polarization switching, attributed to the population imbalance, requires a higher $V_G$ to counteract when $I$ is increased (Fig. 3b).

Our main discovery emerges when $V_G$ is positive. In contrast to the p-doping case, we see the following observations in Fig. 3. The polarization switching displays a weak $I$-dependence for the n-doping case. Interestingly, the threshold voltage is close to the Mott insulating state ($\nu = 1$) where electrons occupy the moiré unit cell in a one-to-one correspondence triggered by on-site Coulomb repulsion ($U_{e-e}$). The polarization switching (from the co-circularly polarized to the linearly-polarized PL) cannot be understood by the valley-flip scattering; it does not add any additional polarization switching mechanisms in the n-doping case[18]. Rather, we consider two scenarios when additional electrons are doped into the lattices, i.e., for $\nu \geq 1$. First, the extra electron may occupy the same orbital as the former one with an opposite spin, resulting in both electrons occupying the same site. Second, when $U_{e-e}$ exceeds the energy differences between two moiré local minimums ($\Delta E_g$), the extra electron avoids double-occupancy and resides at the second local minimum site. In this case, the correlation lead to a charge transfer insulating state and we put the strong correlation-induced moiré exciton site transition as the key contributor for polarization switching, which is in line with the fact that the $WSe_2/WS_2$ heterobilayer is indeed a charge transfer insulator[11,21,22] for $\nu > 1$.

## Ultrafast nonequilibrium optical spectroscopy

To gain dynamic information on the polarization switching phenomena, we have explored the photoinduced nonequilibrium exciton dynamics, by employing ultrafast pump-probe spectroscopy. The time-resolved experiments have been conducted using a 250 kHz Ti:sapphire regenerative amplifier system (Coherent RegA 9040). The pump pulses, whose photon energy is tunable from 1.65 eV to 2.3 eV, are generated by using an Optical Parametric Amplifier (Coherent OPA 9450). Meanwhile, the white-light probe pulses are generated by focusing the 50-fs, 1.55-eV pulse onto a 0.5-mm thick sapphire disk. Typical excitation fluences $F$ ($7 \sim 70\mu J/cm^2$) employed in our experiment yield 2D carrier densities in the range from $10^{11}$ to $3 \times 10^{12} cm^{-2}$ which are below the 2D Mott density of TMDs ($\sim 10^{13} cm^{-2}$)[23–27], suggesting that excitons can be formed from the photoexcitation without forming the electron-hole plasma. First, we monitor the K valley moiré intralayer exciton, i.e. the one at 1.684 eV ($X_1$), under co- and cross-circularly polarized photoexcitation at 1.75 eV (see the inset of Fig. 4a). The contour plots in Figs. 4a and 4b show the measured temporal evolution of the differential reflectance $\Delta R/R_0$ of the moiré intralayer exciton, and the corresponding line-cut spectra at selected pump-probe time delay $\Delta t$ are shown in Fig. 4c. A distinct shape of the $\Delta R/R_0$ spectrum is apparent when comparing the co- and cross-circularly polarized data. Clear exciton-resonance blueshift is observed for the case of the co-circularly polarized pump-probe experiment, whereas the cross-circularly polarized pump-probe data display a redshift in the exciton resonance. The measured $\Delta R/R_0$ spectra remain such features over 3 ns except at the early $\Delta t < 0.1$ ns.

We now discuss the possible scenarios behind the observed spectral transients for early- $\Delta t < 0.1$ ns and long-delay $\Delta t > 0.1$ ns dynamics. For the early-delay dynamics, there are several carrier scattering processes involved such as hot-phonons, carrier-carrier interactions and interlayer exciton transients. Upon ultra-short pulse excitation, high-temperature carriers are generated, experiencing

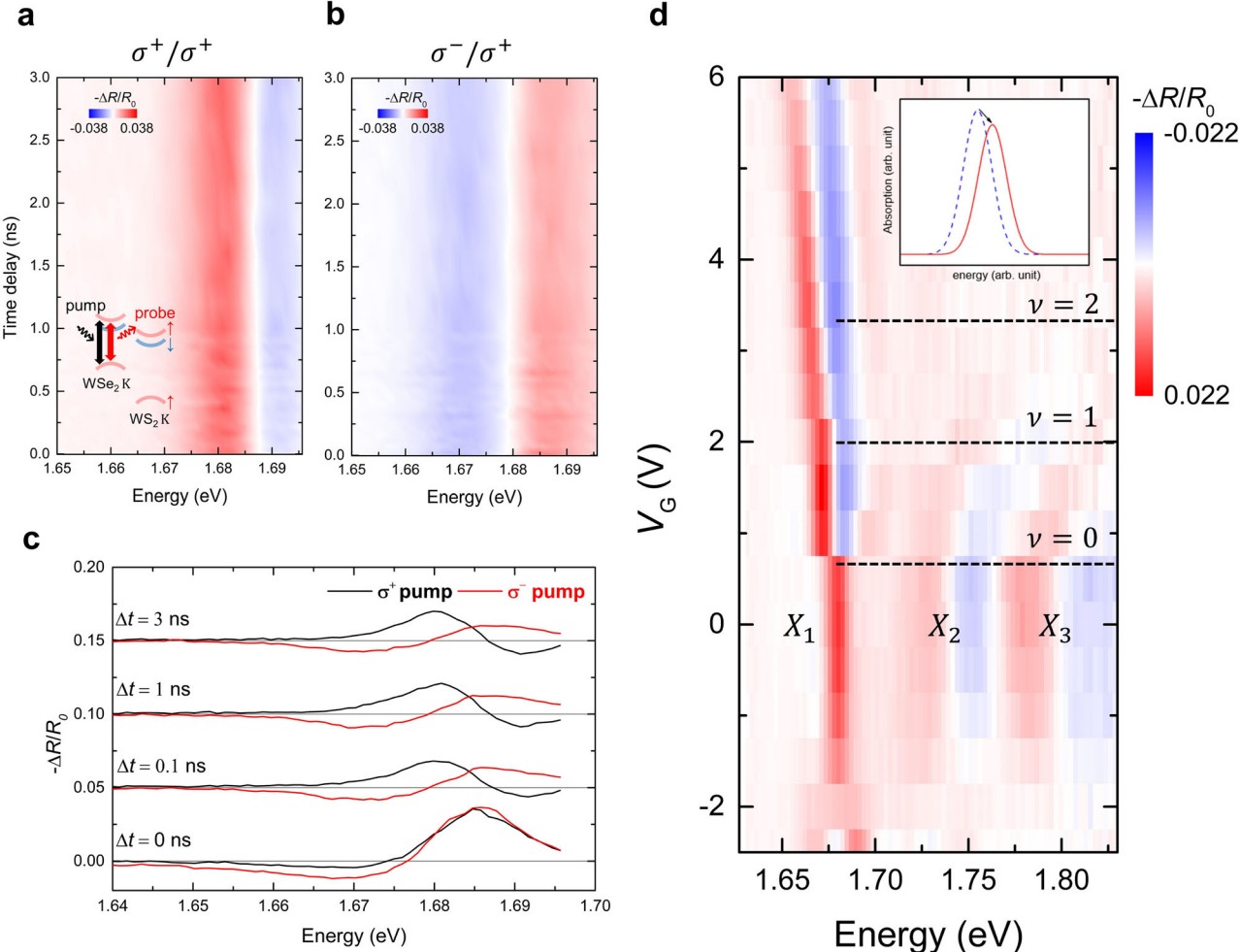

**Fig. 4 | Strongly enhanced Pauli blocking of X₁ in the n-doping regime.** $\Delta R/R_0$ spectra are shown under the pump excitation of $\hbar\omega = 1.75\,\text{eV}$ (**a**, inset) when $V_G = 0\,\text{V}$. **a** Co-circularly polarized and **b** cross-circularly polarized $\Delta R/R_0$, respectively. **c** $\Delta R/R_0$ at selected $\Delta t$ are shown. When $\Delta t = 0$ ns, $\Delta R/R_0$ are similar when the pump and the probe pulses are co-circularly (black solid) and cross-circularly polarized (red solid), indicating the valley-independent hot-carrier dynamics near zero-time delay. However, when $\Delta t > 0.1$ns, the $\Delta R/R_0$ spectra show the valley-dependent interlayer exciton dynamics. **d** $V_G$-dependent $\Delta R/R_0$ measured at $\Delta t = 2$ns for the device R1. Prominent blueshift is observed when $\nu > 0$ (inset). The pump photon energy is 2.0 eV and $F$ is $12\,\mu\text{J/cm}^2$. Corresponding photoexcited exciton density, estimated using a transfer matrix method (Supplementary Note 3), is $2.79 \times 10^{12}\,\text{cm}^{-2}$. This is the same order of moiré density $n_0 = 2.19 \times 10^{12}\,\text{cm}^{-2}$.

thermalization relaxations through hot-phonon and carrier-carrier interaction in a few hundred femtoseconds[24,28]. In addition, ultrafast charge transfer in a type II band alignment occurs within a few hundred femtoseconds, whose time scales depend on the stacked materials[29,30]. Subsequently, the intralayer exciton is formed within a few picoseconds in typical monolayer TMDs[31–33], and the interlayer exciton is formed within 1 ps[34]. Thus, until a few picoseconds, the spectral transients are influenced by the combined effects of the photoexcited hot-carriers, hot-phonons and the interlayer exciton formation. However, because the interlayer excitons exhibit long-lived lifetimes (~1.8 ns)[35] and ultralong valley lifetimes (~40 μs)[36,37], the longer time delay (e.g., $\Delta t = 2$ ns) is appropriate for studying the dynamics of interlayer exciton without considering the early transient effects. We first focus our discussions on the longer time delays to elucidate the interlayer exciton site transition dynamics. Subsequently, we address the early time delays, where the relaxation dynamics is primarily dominated by hot-phonon cooling processes.

The co-circularly polarized exciton resonance displays a blueshift at all the long-time delays (Fig. 4a). This can be attributed to the Burstein-Moss effect-induced Pauli blocking of the band-edge transition[38,39]. The ultralong valley lifetime[36,37] in WSe₂/WS₂ ensures that the co-circularly polarized signal is primarily influenced by this

effect, because the photoexcited carriers tend to remain within the same valley where they were initially excited. Conversely, for the cross-circularly polarized case, the Pauli blocking subdues significantly because most of the carriers reside in the opposite valley. Of course, the valley-independent effects, such as bandgap renormalization[31,38] or lattice heating[40], can still induce the absorption changes. These factors lead to a redshift in the resonance energy, consistent with our results (Fig. 4b).

Novel aspects of our study are the $\nu$-depedent nonequilibrium $\Delta R/R_0$ spectra. Supplementary Fig. 5a depicts the $\Delta R/R_0$ spectra for different filling, $\nu$ from −2 to +2, where we have tuned the pump-photon energy of 2.0 eV, near the resonance of the WS₂ A exciton. Akin to Figs. 4a and 4b, no significant spectral re-shaping is observed in the $\Delta R/R_0$ spectra after a sufficient time delay of around 0.1 ns. This confirms that our data represent the exclusive dynamics of the interlayer excitons. A comparison between $\nu = 0$ and $\nu = 2$ case shows that the latter exhibits a more pronounced blueshift than the former. We observe similar behaviors across various $F$'s with a lower photoexcitation energy of 1.75 eV (Supplementary Fig. 6). Figure 4d shows the $\nu$-dependent interlayer exciton $\Delta R/R_0$ spectra at $\Delta t = 2$ns. The $\Delta R/R_0$ spectral transients of $X_1$ start to exhibit an abrupt change and a strong blueshift when $\nu$ is above zero filling, i.e. $V_G$ is larger than 1 V. On

the other hand, signals from $X_2$ and $X_3$ exhibit blueshifts at $V_G = 0V$, and they eventually subside without noticeable spectral changes even when $\nu$ is larger than the zero ($V_G > 1V$) (see Supplementary Note 1 for the detailed discussion about the $X_1$, $X_2$, and $X_3$ dynamics and Supplementary Fig. 7 for same contour plots resulted from device R3 across various $F$'s).

The enhanced blueshift observed near the Mott insulating state ($\nu = 1$) can be understood on the basis of the large-scale first-principles calculations. Recent theory and experiment have demonstrated that the lowest energy moiré exciton ($X_1$ in our case) is a Wannier-type exciton, featuring tightly bounded electrons and holes localized at the $R_h^h$ site in the moiré supercell[13]. In contrast, the highest energy moiré exciton ($X_3$ in our case) is characterized as an intralayer charge-transfer exciton with electrons and holes spatially separated. It has also been pointed out that the doped electron occupies the $R_h^X$ site in the moiré supercell, which is the same site of interlayer excitons in the neutral regime. When n-doping is introduced, it naturally invokes a Coulomb repulsion ($U_{ex-e}$) between the doped electrons and the photoexcited interlayer excitons, causing the latter to dynamically shift to a more energetically favorable second local minimum site, i.e. $R_h^h$. In Supplementary Note 2, we discuss the exciton site transition by comparing energy differences of $\Delta E_g$ and $U_{ex-e}$. Within the framework of a site transition in a moiré supercell, the measured $\Delta R/R_0$ spectra at $\nu \geq 1$ clearly represent the avoided double occupancy of Pauli blocking experienced by the localized exciton $X_1$. In fact, the observed polarization switching from the circularly polarized PL to the linearly polarized one is consistent with the standard optical selection rule of the interlayer exciton PL (Supplementary Table 1). In Supplementary Fig. 8, we have shown extensive temperature dependent $\Delta R/R_0$ spectra. The measured critical temperature for the enhanced Pauli blocking is comparable with the thermal-activation temperature of Mott-Hubbard gap in WSe$_2$/WS$_2$. This implies that such phenomena originate from the correlation-driven effects.

In the case of H-stacking, the moiré potential exhibits different potential landscapes in comparison to the R-stacking. This distinction arises from the differences in atomic configuration stemming from different twist angles. Here, the excitons and doped electrons are localized at a different site in contrast to the R-stacking. While recent studies[10,18] have investigated the site of interlayer excitons and doped electrons in the H-stacking, the site of the moiré intralayer exciton (for the case of H-stacking) has not been clearly resolved. Nevertheless, a natural expectation is that the enhanced Pauli blocking shall result in a blueshift of the spectral transient if interlayer excitons and moiré intralayer excitons occupy the same site. If not, we may observe a spectral redshift due to bandgap renormalization or lattice heating. The transient redshift or blueshift is mainly determined by how these exciton wavefunctions are spread out in a moiré supercell. We substantiate this scenario by performing the same experiment on the device H1, and have found that the competing processes between the blueshift and the redshift mechanisms yielded a modest blueshift without abrupt changes (Supplementary Fig. 10b). While a comprehensive theoretical work is necessary, the spectral transients without the abrupt changes indicate the absence of polarization switching in the device H1 with no moiré exciton site transition.

To focus on the population dynamics while excluding the effect of spectral shift, we perform spectral integration on the differential reflectance $\Delta R/R_0$[41,42] (Fig. 5a). We use a biexponential fit function to examine the decay of population (see Supplementary Note 4 for more details). The fast and slow components of decaying dynamics are denoted as $\tau_{fast}$ and $\tau_{slow}$, respectively. We choose an excitation energy of 1.75 eV, which is in resonance with $X_2$. We investigate the $F$ dependence of $\tau_{fast}$ and $\tau_{slow}$ for various fillings $\nu$ from -2 to 2. Upon a relatively small $F$ of 7.2 μJ/cm², we extract $\tau_{fast}$ of $0.44 \pm 0.2$ ps when $V_G = 0V$ and (Fig. 5b). With a high $F$ of 36.2 μJ/cm², $\tau_{fast}$ increases to a

value of $0.49 \pm 0.04$ ps. While a similar $F$ dependence is observed for each filling, the noticeable changes are seen at $V_G = 4V$, and the $F$ dependence is almost negligible at $V_G = 2V$. The observed behavior is consistent with the hot phonon effect[24,43,44], where a substantial phonon population is induced by the elevated lattice temperature, which prevent the photoexcited carriers from being cooled via phonon emission. This effect contradicts with the carrier-carrier scattering process, where the sub-picosecond relaxation is known to become faster with increasing $F$. In prior studies[24,45], $A_{1g}$ and/or $E_{2g}^1$ optical phonons are proposed as the candidates in the cooling process within a time scale of 0.5 ps. Meanwhile, the modulation of optical phonons frequency and intensity through the carrier doping is reported by previous studies[46-48], which may lead to different values of $\tau_{fast}$ and the dependency in various $\nu$'s.

The slow decay component $\tau_{slow}$ is on the order of ns, which represent an effective lifetime of interlayer excitons in R-stacked WSe$_2$/WS$_2$ heterobilayers (Fig. 5c). When $V_G = 0V$ and $F = 7.2$ μJ/cm², the measured $\tau_{slow}$ is a value of $3.38 \pm 0.33$ ns. This time constant does not match with recent studies[49,50] of reporting relatively short interlayer exciton lifetime of ~1 ns for R-stacked WSe$_2$/WS$_2$ heterostructures. The discrepancy may arise because our experiments were performed at a temperature of 4 K, minimizing the non-radiative recombination[49,51]. With increasing $V_G$, the extracted $\tau_{slow}$ is $2.24 \pm 0.19$ ns at $V_G = 2V$ and $2.25 \pm 0.23$ ns $V_G = 4V$, respectively, which are further decreased compared to the neutral regime. This is consistent with the study[11] reporting a decrease in the radiative lifetime of interlayer exciton in WSe$_2$/WS$_2$ when more n-doping is introduced. Interestingly, no significant $F$ dependence is observed. It suggests a negligible exciton-exciton annihilation process due to the minimal density of photoexcited interlayer exciton. However, in the case of p-doping, $\tau_{slow}$ not only strongly depends on $F$, but also exhibits opposite $F$ dependencies for $V_G = -2.3V$ and $V_G = -3.9V$. Assuming the radiative decay rate remains relatively unchanged throughout the p-doping[11], we can infer that the non-radiative decay and the related processes undergo significant changes, particularly due to gap opening at each insulating state, as previously observed in WSe$_2$/MoS$_2$ heterobilayers[52].

We now turn to the rise dynamics of the population. Note that our independent measurement shows that a temporal resolution of our pump-probe signal is limited to 132 fs (Supplementary Fig. 11). To monitor and quantify the site transition time (i.e., $\tau_{site}$ in Fig. 5d), we focus on the rise dynamics of the $\Delta R/R_0$ signals for $X_1$ exciton under the photoexcitation on $X_2$. When $V_G = 0V$, $\tau_{site}$ is measured to be $267 \pm 58$ fs (Fig. 5d). We note that this site transition dynamics precedes the charge transfer dynamics, which occurs within $383 \pm 10$ fs (see Supplementary Fig. 12 for full $V_G$-dependent charge transfer dynamics). Thus, the dynamics of electrons and holes cannot be distinguished in this case, because it occurs before the separation of electrons and holes in different layers. Upon p-doping, $\tau_{site}$ slightly increases to $339 \pm 87$ fs at $V_G = -3.9V$. Conversely, upon n-doping, it significantly decreases, reaching our temporal resolution limit (represented by the dashed line) at $V_G = 2V$ (Fig. 5e). These trends are consistently observed with various $F$ (Supplementary Fig. 13). Recent theoretical and experimental studies[17,53] suggest that the electrostatic doping can alter the landscape of moiré potentials. The asymmetric dependency of $\tau_{site}$ on doping may reveal this signature of site-dependent dynamics of carriers. One possible scenario is that, with increasing the n-doping, the site of ground exciton state experiences a dynamic transition from $R_h^X$ to $R_h^h$, leading to the efficient carrier scattering into the $R_h^h$ site, which is the energetically favorable site. In fact, this is in line with our finding of exciton site transition that occurs exclusively in the n-doping, not in the case of p-doping. However, more theoretical studies for R-stacked WSe$_2$/WS$_2$ heterobilayers are required to understand the experimental results, which is beyond the scope of the current study.

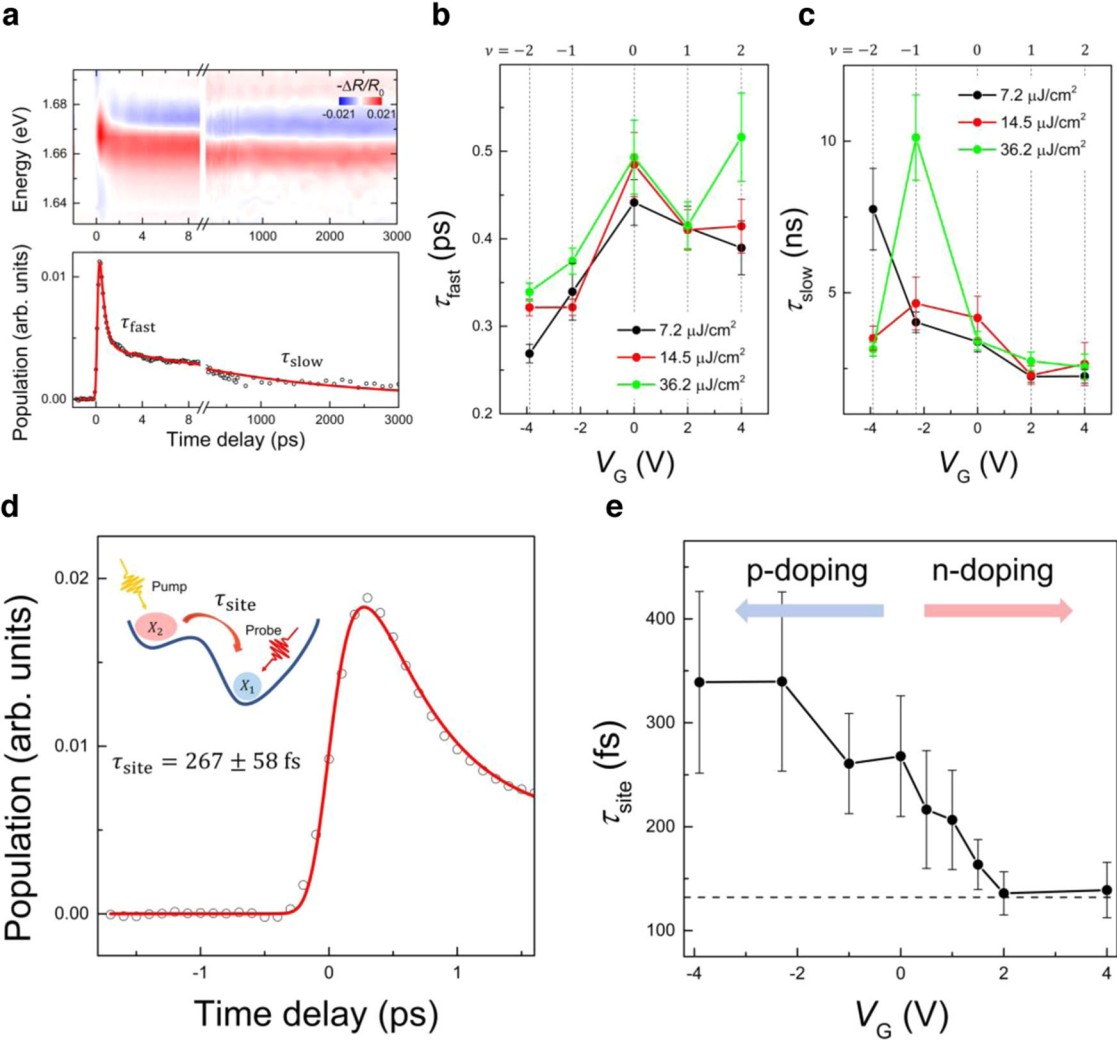

**Fig. 5 | $V_G$-dependent nonequilibrium decaying and rising dynamics. a** The measured early- and long-delay $\Delta R/R_0$ spectra (top) and the spectrally integrated $\Delta R/R_0$ (bottom) as a function of time delay at $V_G = 4\,V$, showing a temporal evolution of the carrier population. **b, c** Fast ($\tau_{fast}$) and slow ($\tau_{slow}$) decay time constants as a function of $V_G$ under three different $F$'s. The black, red and green circle correspond to the different $F$'s of 7.2, 14.5, and 36.2 $\mu J/cm^2$, respectively. Each integer filling $\nu$ from −2 to 2 is represented by dashed lines. We propose that the observed decaying dynamics of $\tau_{fast}$ shows the carrier cooling dominated by hot phonon effects. Meanwhile, $\tau_{slow}$ shows 4-order longer dynamics than $\tau_{fast}$, which is directly

related to the lifetime of interlayer excitons. Distinct $F$ dependence is evident when $V_G < 0\,V$, while no significant change is observed when $V_G \geq 0\,V$. **d** Temporal evolution of carrier population (open black circles). The red line is the fit to the data. We correlate the rise dynamics of $X_1$ (1.68 eV) under photoexcitation resonant with $X_2$ (1.75 eV) to the site transition dynamics ($\tau_{site}$). **e** $V_G$-dependent evolution of $\tau_{site}$ (black circles) with $F$ of 14.5 $\mu J/cm^2$. A dashed black line indicates the temporal resolution limit of 132 fs (Supplementary Fig. 11). Vertical error bars in **b**, **c**, and **e** are obtained from the fits.

We have observed the correlation-driven nonequilibrium exciton site transition in near-zero angle-aligned $WSe_2/WS_2$ heterobilayers. Beyond the conventional steady-state optical measurements, the site-dependent exciton correlations have been evidenced by the ultrafast optical spectroscopy, whose effects enable to explain the polarization switching and strongly enhanced Pauli blocking when the doping is close to the Mott insulating regime. These findings, with the long-lived interlayer excitons, are expected to facilitate future researches into the unexplored nonequilibrium correlations in TMDs heterobilayers such as temporal dynamics of Bosonic correlated states and exciton-phonon interaction perturbed by the correlated electrons, which have been challenging to explore using conventional optical spectroscopy.

## Methods
### Sample fabrication
We exfoliate monolayers of $WSe_2$ and $WS_2$ from bulk crystals (HQ graphene) and transfer them onto a $SiO_2/Si$ substrate. We then

conduct polarization-resolved second harmonic generation (SHG) measurements to identify the crystal orientation. Next, we transfer the $WSe_2$ and $WS_2$ monolayers onto a polyethylene terephthalate (PET) stamp. We assemble a near zero-degree-twist-angle heterostructure using these materials, with few-layer graphite (FLG) acting as the electrical contact. This heterostructure is enclosed by the top and bottom hexagonal boron nitride (hBN) dielectric layers, whose thickness are 17 nm and 25 nm confirmed by atomic force microscopy (AFM). High-quality single-crystal hBN has been provided by Advanced Materials Laboratory (National Institute for Materials Science, Japan). For our experimental setup, we create a back gate using a standard e-beam deposition system, comprising a 5 nm thick Ti layer and a 45 nm thick Au layer. Following the fabrication process, we have performed additional polarization-resolved SHG measurements. We use these measurements to precisely determine the twist angle in the monolayer regions of the sample and to distinguish between samples with near-zero and near-sixty-degree twist angles in the heterostructure region.

## Optical measurements

A halogen lamp (Thorlabs OSL2) was used as a white-light source for the reflection contrast (RC) measurements. The output of the lamp was passed through a single-mode fiber (Thorlabs P1-630P-FC) and collimated by a triplet collimator (Thorlabs TC18FC). Then the beam was focused onto the sample with a 50× long working distance objective (Mitutoyo Plan Apo SL, NA = 0.42). The beam diameter on the sample was about 1μm, with a beam power of less than 5nW. The reflected beam from the sample was collected using the same objective and then dispersed using a diffraction grating with 150 grooves per millimeter. This dispersed beam was then detected using a charge-coupled device camera (Oxford Newton 970 EMCCD). The RC spectrum, defined as $RC \equiv (R - R')/R'$, was obtained by comparing the reflected light intensity from the sample ($R$) to that of a spectrum ($R'$) from the background. The measurement sensitivity is about 0.1%. For the PL measurements, we used a 632.8 nm diode laser, which shares the same optical path of the RC measurements. A 633 nm notch filter (Thorlabs NF633-25) was used to spectrally filter out the laser line.

Ultrafast pump-probe spectroscopy was conducted using a 250 kHz Ti:sapphire regenerative amplifier system (Coherent RegA 9040). In this system, the 1.55 eV pulses that have a duration of 50 femtoseconds were generated and then were separated into two paths using a beam splitter for the pump-probe spectroscopy. The pump pulses, with variable photon energy ranging from 1.65 eV to 2.3 eV, were produced through Optical Parametric Amplification (Coherent OPA 9450). Meanwhile, the white-light probe pulses were generated by focusing the 1.55 eV pulse on a 0.5-mm thick sapphire disk. Prism pairs (Thorlabs SF10) were placed in both paths to compensate the dispersion introduced by dispersive optical elements. A dual-slotted chopper (Stanford Research Systems SR 540) was utilized to simultaneously record the reflectance without pump ($R_0$) and the differential reflectance ($\Delta R$) with pump via the dual lock-in detection technique (Stanford Research Systems SR 830). The probe beam reflected from the sample was passed through a monochromator (Dongwoo optron MonoRa 512i) before reaching an avalanche photodiode (Thorlabs APD410A). We recorded $\Delta R$ of the probe beam while varying the pump-probe time delay ($\Delta t$) and the probe wavelength. Laser polarization and power were controlled using a proper set of waveplates (AQWP05M, AHWP05M-580, 10RP52-2) and a linear polarizer (GL10-A, GL10-B, 10GT04) depending on the laser wavelength. The entire experiment was performed in a closed-cycle Montana cryostat (Cryostation s50) while keeping the base temperature of 4 K.

## Determination of moiré density

The moiré density $n_0$ is given by $n_0 = 2/(\sqrt{3}a_M^2)$ for a triangular superlattice and the filling factor $\nu$ is given by $\nu = n/n_0$. Here $a_M$ is the moiré superlattice constant which is determined by the twist angle ($\theta$) and lattice mismatch $\delta = (a_{Se} - a_S)/a_{Se}$ between WSe$_2$ and WS$_2$, where $a_{Se}$ of 0.328 nm is the lattice constant of WSe$_2$ and $a_S$ of 0.315 nm is the one of WS$_2$. We obtain $a_M \sim 7.53$nm by using the relation $a_M = a_{Se}/\sqrt{\delta^2 + \theta^2}$. Corresponding moiré density $n_0$ is $2.19 \times 10^{12}$ cm$^{-2}$ for $\theta = 1.2°$.

## Data availability

The data that support the findings of this study are available from the corresponding author upon request. The full set of pump-probe data generated in this study are provided in the Supplementary Information file.

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

## Acknowledgements

This research was supported by the National Research Foundation of Korea (NRF) through the government of Korea (Grant No. 2021R1A2C3005905), Scalable Quantum Computer Technology Platform Center (Grant No. 2019R1A5A1027055), the Institute for Basic Science (IBS) in Korea (Grant No. IBS-R034-D1), Global Research Development Center (GRDC) Cooperative Hub Program through the National Research Foundation of Korea (NRF) funded by the Ministry of Science and ICT (MSIT) (Grant No. RS-2023-00258359), and the core center program (2021R1A6C101B418) by the Ministry of Education.

## Author contributions

Jinjae Kim, J.P., and Hyojin Choi fabricated samples. T.K., S.C. and Y.L. performed the device characteristics examination. K.W. and T.T. provided high-quality hBN crystal. Jinjae Kim and J.P. performed the measurements. Jinjae Kim, J.P., Jonghwan Kim, M.-H.J. and Hyunyong Choi performed data analysis and discussed the results. Hyunyong Choi supervised the project. Jinjae Kim and J.P. wrote the manuscript with input from all co-authors.

## Competing interests

The authors declare no competing interests.
