## [Peer Review File · Nature Communications]

REVIEWER COMMENTS

Reviewer #1 (Remarks to the Author):

Kim et al. report on the correlation-driven non-equilibrium exciton site transition within a WSe₂/WS₂ moiré supercell. The authors use static spectroscopy in order to characterize the optical response of the moiré heterostructure as a function of the filling factor. Based on the title, abstract and introduction, I expect that the authors mainly aim to understand the non-equilibrium dynamics of excitons in the presence of correlated interactions, which is why they perform a pump-probe optical spectroscopy.

While the static optical response of the TMD heterostructure and the impact of the moiré potential is reported by many authors, I agree with the authors of the present manuscript that it is a highly interesting research direction to study the non-equilibrium response of the heterostructure for different filling factors. Hence, I think that that the research of the authors around Fig.4 (extended Fig. 5) is highly interesting for different communities and thus might be interesting for Nature Communications.

However, in the current form, I think the manuscript does not provide substantial new insights with regard to the 'non-equilibrium' response.

In this manner, I would like to ask several questions:

- I do not understand the wording of the second sentence of the abstract. To my understanding, the exciton resonance and the dynamic interactions are site-dependent.
- Labels d-f are missing in Fig.2.
- On page 8, the authors argue why it is reasonable to only show and discuss the long-time dynamics (>0.1 ns). From the perspective of the non-equilibrium response of the TMD heterostructure to an optical excitation, I would think that especially the early femto- to picosecond dynamics is interesting. In how far is the interlayer charge transfer and the ILX formation dynamics dependent on the filling factor? In the last years, there has been huge progress on the understanding of the charge-transfer dynamics based on theory, trARPES and tr-reflectivity experiments. If the authors are interested in the non-equilibrium response, I think it would be highly important to discuss such data in the manuscript.
- Based on Fig.4a-c, the authors give a first overview of the time-resolved data and then explain the major features in direct comparison to literature. In line 174, the authors state that the novel aspect of their study is the filling dependent time-resolved reflectivity data. Also from my perspective, these

data are the most exciting experimental achievement. Why don't the authors show such data in the main text? I would be highly interested to understand details of the differing dynamics from extended Fig. 5 (and the femtosecond dynamics).

- For the filling factor dependent time-resolved experiments, the pump photon energy is tuned to 2.0 eV (extended Fig.5). Why is the photon energy chosen to be different as compared to Fig. 4 (1.75eV). For the 1.75eV, I can understand that the idea is to only excite WSe₂... this simplifies the possible scattering dynamics of excitons and charge carriers. Why is the photon energy of 2.0eV chosen for the filling dependent experiment? My guess is that the filling factor dependent dynamics is different for different exciton photon energies?

- Starting line 176, the authors state that the early time dynamics is not dependent on the filling factor. I think it would be important and highly interesting to show such data. Why is the charge transfer and ILX formation dynamics not dependent on the filling factor?

- In how are is the early and long-time dynamics pump power dependent (i.e., dependent on the exciton density)?

- Are similar data as shown in extended Fig. 5 as well observed if the sample temperature is increased? Is there a critical temperature to observe such dynamics

- Do I understand correctly that the time-dependent blue-/redshift is a consequence of the Coulomb repulsion of the doped electrons and the photoexcited ILX? If so, can this be made more clear by labelling spectral features or trends in the figures? In how are does the time-scale of the blue-/redshift agree with this explanation? Hence, in how are is it a non-equilibrium response?

Reviewer #2 (Remarks to the Author):

The manuscript by Kim et al reports a study on the polarization resolved optical spectra of twisted WSe₂/WS₂ hetero-stacking. The authors observed polarization switching and absorption edge shifts in near-zero angle aligned WSe₂/WS₂ heterobilayers. They attributed the observation to correlation-driven effects. Overall, the authors presented certain new findings (by both of CW and ultrafast laser excited methods), and I appreciate the experimental data quality; but in my opinion it is too "technical" to falls into a specified area, which may not attracts broad interesting in the community. Although it is well organized and clearly written, the manuscript calls for in-depth analysis or insightful physics for publishing in Nature Communications.

My major concern is about the working mechanism. The authors stated that the polarization switching results from strong correlation induced exciton site transition. It is not clear to me what is the role played by the correlation effect. How does it affect the site transition in real space? And why

does it occur at/around the “Mott insulator” state? It would be necessary to clearly point out the physics picture behind.

Another flaw of this manuscript is about the filling factor calculation. Generally, filling factors for a Moire hetero-stacking are calculated from the area of Moire super cell. It looks that the authors assigned the filling factors according to the optical spectrum. It is quite strange that the filling factors are not equally spaced. The assignment is somehow arbitrary. And it hinders the validity of the conclusion.

And in addition, a minor point: labels of d-f are missing in figure 2.

Point-by-point responses to the issues raised by the reviewers

General remarks and comments of Reviewer 1:

Kim et al. report on the correlation-driven non-equilibrium exciton site transition within a WSe₂/WS₂ moiré supercell. The authors use static spectroscopy in order to characterize the optical response of the moiré heterostructure as a function of the filling factor. Based on the title, abstract and introduction, I expect that the authors mainly aim to understand the non-equilibrium dynamics of excitons in the presence of correlated interactions, which is why they perform a pump-probe optical spectroscopy.

While the static optical response of the TMD heterostructure and the impact of the moiré potential is reported by many authors, I agree with the authors of the present manuscript that it is a highly interesting research direction to study the non-equilibrium response of the heterostructure for different filling factors. Hence, I think that that the research of the authors around Fig.4 (extended Fig. 5) is highly interesting for different communities and thus might be interesting for Nature Communications.

However, in the current form, I think the manuscript does not provide substantial new insights with regard to the ‘non-equilibrium’ response.

In this manner, I would like to ask several questions:

Response 1:

We thank Reviewer 1 for taking time to review our manuscript. We appreciate the Reviewer’s agreement of the significance of our research direction, particularly in studying the nonequilibrium response of the heterostructure for different filling factors. We agree with the comments that providing more detailed studies on the nonequilibrium pump-probe spectroscopy would be highly interesting to the communities of twisted TMDs. Following Reviewer’s fruitful advice, we have carefully considered the points/issues raised by Reviewer 1 and have thoroughly revised our manuscript by providing additional nonequilibrium experimental data to support our main idea. We also made corrections on some expressions, which caused misunderstandings in the original manuscript.

While we prepare the responses, we fabricated an additional R-stacked device. We explored the early dynamics of the moiré excitons on a time scale of a few hundreds of femtoseconds (Comments 1-3, 4, 6, 7). Indeed, we have found intriguing phenomena that are consistent with the concept of site-dependent dynamics. The new findings are presented in Fig. 5 of the revised manuscript, and we also have added a qualitative explanation on the newly measured dynamics.

Below we present our point-by-point responses to the comments raised by Reviewer 1.

Comments 1-1: I do not understand the wording of the second sentence of the abstract. To my understanding, the exciton resonance and the dynamic interactions are site-dependent.

Response 1-1: We appreciate the Reviewer's attention on the details of the second sentence. We fully agree that it needs to be clarified, and we also have acknowledged the confusion arising from the wording. To provide a clearer explanation, our intent is consistent with the Reviewer's understanding: exciton resonances and the dynamic interactions between excitons and the doped electrons are site-dependent within a moiré supercell. In the revised manuscript, we have corrected the sentence as below.

Before: Whereas the spatially modulated potentials evoke that the exciton resonances are distinct depending on a site in a moiré supercell, it is presumed that the dynamic interactions with electrons were to be site-dependent.

After: Whereas the spatially modulated potentials evoke that the exciton resonances are distinct depending on a site in a moiré supercell, there have been no clear demonstration how the moiré excitons trapped in different sites dynamically interact with the doped carriers; so far the exciton-electron dynamic interactions were presumed to be site-dependent.

Comments 1-2: Labels d-f are missing in Fig.2.

Response 1-2: We sincerely apologize for any confusion due to the missing labels in Fig. 2. We have immediately corrected this mistake and conducted a thorough revision of the manuscript to prevent any recurrence of such errors again.

Comments 1-3: On page 8, the authors argue why it is reasonable to only show and discuss the long-time dynamics (>0.1 ns). From the perspective of the non-equilibrium response of the TMD heterostructure to an optical excitation, I would think that especially the early femto- to picosecond dynamics is interesting. In how far is the interlayer charge transfer and the ILX formation dynamics dependent on the filling factor? In the last years, there has been huge progress on the understanding of the charge-transfer dynamics based on theory, trARPES and tr-reflectivity experiments. If the authors are interested in the non-equilibrium response, I think it would be highly important to discuss such data in the manuscript.

Response 1-3: We thank Reviewer 1 for his/her constructive comments. As highlighted by Reviewer 1, there have been notable progresses on the nonequilibrium dynamics in the TMD heterostructures, which include, for example, the charge-transfer dynamics [Nature **603**, 247-252 (2022); Nature **608**, 499-503 (2022)] and the interlayer exciton formation dynamics [Nat. Commun. **14**, 7273 (2023)]. Existing experimental studies, however, have reported somewhat inconsistent results of the charge transfer dynamics. Moreover, they are particularly dependent of the stacked materials. For example, the charge transfer in MoS₂/WSe₂ heterostructures has been reported to be faster than the experimental resolution of 40 fs independent of the twist angles [Nano Lett. **17**, 3591-3598 (2017)]. In contrast, WS₂/WSe₂ exhibits much slower charge transfer times in the range of a few hundred femtoseconds, with a substantial increase to 1.2 ps for larger rotational mismatch [Nat. Mater. **18**, 691-696 (2019)]. In this context, it is necessary to examine both experimental and theoretical studies for the specific material combinations, considering their distinct microscopic structures. To the best of our knowledge, there are no experimental reports concerning the filling-dependent charge transfer dynamics in any transition metal dichalcogenides heterostructures. Thus, we agree with the importance emphasized by Reviewer 1 regarding the demonstration of filling-dependent charge transfer dynamics. Below we present our experimental results measured on a newly fabricated R-stacked device.

For the additional experiments, we have used the pump pulse of 1.68 eV photon energy to resonantly excite the intralayer A-excitons in WSe₂. The results are summarized in Supplementary Fig. 12 in the revised manuscript. We extract the charge transfer time (i.e., τ_{transfer} in Fig. R1a) by monitoring the rise dynamics of WS₂ A-exciton resonances (2.0 eV) with varying the gate voltage V_G . At $V_G = 0$ V, τ_{transfer} is estimated to be 383 ± 10 fs. This time remains relatively constant or slightly decreases at the Mott insulating state ($\nu = \pm 1$ in Fig. R1b). In contrast, at $V_G = -3.9$ V and $V_G = 4$ V, τ_{transfer} becomes longer, reaching 629 ± 28 fs and 487 ± 36 fs,

respectively. Here we wish to make a statement that despite the increased doping leads to an increased τ_{transfer} , the time scale remains below 1 ps. In Fig. 4d, because our discussion has focused on the interlayer exciton site transition at the long delay time (e.g. $\Delta t = 2$ ns), it is appropriate for studying the dynamics of interlayer exciton without considering the early transient effects. Nevertheless, the question of site-dependent “nonequilibrium” femtosecond dynamics still remain infancy. Below, we explore and measure another aspect of femtosecond nonequilibrium dynamics, which we refer to as the site transition time. Please note that the discussions shown below are the same paragraph in the revised manuscript (page 13-14).

Note that our independent measurement shows that a temporal resolution of our pump-probe signal is limited to 132 fs (Fig. R2). To monitor and quantify the site transition time (i.e., τ_{site} in Fig. R3a), we focus on the rise dynamics of the $\Delta R/R_0$ signals for X_1 exciton under the photoexcitation on X_2 . When $V_G = 0$ V, τ_{site} is measured to be 267 ± 58 fs (Fig. R3a). We note that this site transition dynamics precedes the charge transfer dynamics. Thus, the dynamics of electrons and holes cannot be distinguished in this case, because it occurs before the separation of electrons and holes in different layers. Upon p-doping, τ_{site} slightly increases to 339 ± 87 fs at $V_G = -3.9$ V. Conversely, upon n-doping, it significantly decreases, reaching our temporal resolution limit (represented by the dashed line) at $V_G = 2$ V (Fig. R3b). These trends are consistently observed with various F (Fig. R4). Recent theoretical and experimental studies [Nat. Photon. **13**, 131-136 (2019); npj Comput. Mater. **9**, 8 (2023)] suggest that the electrostatic doping can alter the landscape of moiré potentials. The asymmetric dependency of τ_{site} on doping may reveal this signature of site-dependent dynamics of carriers. One possible scenario is that, with increasing the n-doping, the site of ground exciton state experiences a dynamic transition from R_h^X to R_h^h , leading to the efficient carrier scattering into the R_h^h site, which is the energetically favorable site. In fact, this is in line with our finding of exciton site transition that occurs exclusively in the n-doping, not in the case of p-doping. However, more theoretical studies for R-stacked WSe₂/WS₂ heterobilayers are required to understand the experimental results, which is beyond the scope of the current study.

The above discussions regarding V_G -dependent τ_{site} have been added into the revised manuscript. The results obtained from a newly fabricated device (device R3) are presented in the revised main text (see Fig. 5 and the related discussions). Further details about the fit function utilized are included in Supplementary Note 4 of the revised manuscript.

Figure R1 | V_G -dependent charge transfer dynamics in WSe_2/WS_2 heterobilayers. **a** Temporal evolution of carrier population at the WS_2 A-exciton resonance when pump excitation takes place in the WSe_2/WS_2 heterobilayers. We extract the charge transfer time (τ_{transfer}) by monitoring the rise dynamics of the population signals. τ_{transfer} represents the time for photoexcited electron separation via interfacial charge transfer from WSe_2 to WS_2 due to type-II band alignment (inset). While we estimate $\tau_{\text{transfer}} = 383 \pm 10$ fs at $V_G = 0$ V (black squares), τ_{transfer} becomes longer to 629 ± 28 fs at $V_G = -3.9$ V (red circles). Each solid line corresponds to the biexponential fits. **b** Measured V_G -dependent τ_{transfer} . Note that τ_{transfer} is significantly long when the doping increases, see the cases of $V_G = -3.9$ V and $V_G = 4$ V.

Figure R2 | Degenerate pump-probe measurement at X_1 exciton (1.68 eV). Transient population dynamics is obtained by a spectral integration of $\Delta R/R_0$. We note that the fastest rise time constant (τ_{rise}) is obtained when the exciton is resonantly excited. In this case, τ_{rise} reaches a value of 132 ± 7 fs. We employ linearly cross-polarized pump and probe geometry to filter out the scattering of pump pulse.

Figure R3 | V_G -dependent nonequilibrium rising dynamics. **a** Temporal evolution of carrier population (open black circles). The red line is the fit to the data. We correlate the rise dynamics of X_1 (1.68 eV) under photoexcitation resonant with X_2 (1.75 eV) to the site transition dynamics (τ_{site}). **b** V_G -dependent evolution of τ_{site} (black circles) with F of $14.5 \mu\text{J}/\text{cm}^2$. A dashed black line indicates the temporal resolution limit of 132 fs.

Figure R4 | V_G -dependent τ_{site} for various F . The site transition time (τ_{site}) is measured at various F of 7.2 (black), 14.5 (red), 36.2 $\mu\text{J}/\text{cm}^2$ (green). Across all F , τ_{site} exhibits a decreased feature until it reaches the temporal resolution limit of 132 fs (dashed black line) for $V_G \geq 2$ V. Upon p-doping, τ_{site} not only shows an increased feature compared to the neutral regime, but also becomes longer with higher F .

Comments 1-4: Based on Fig.4a-c, the authors give a first overview of the time-resolved data and then explain the major features in direct comparison to literature. In line 174, the authors state that the novel aspect of their study is the filling dependent time-resolved reflectivity data. Also from my perspective, these data are the most exciting experimental achievement. Why don't the authors show such data in the main text? I would be highly interested to understand details of the differing dynamics from extended Fig. 5 (and the femtosecond dynamics).

Response 1-4: We appreciate Reviewer 1's suggestion on emphasizing the filling-dependent dynamics, i.e. such as shown in Extended Data Fig. 5 (Supplementary Fig. 5 in the revised manuscript). Following Reviewer 1's advice, we have added Figs. 5a-5c in the revised manuscript. Here, we examine not only the filling-dependent but also the fluence-dependent measurement, as pointed in Comments 1-7. We present our responses to Comments 1-4 and 1-7 collectively by providing new data obtained from a newly fabricated device R3. Please note that the discussions shown below are the same paragraph in the revised manuscript (page 12-13).

To focus on the population dynamics while excluding the effect of spectral shift, we perform spectral integration on the differential reflectance $\Delta R/R_0$ [Nat. Commun. **9**, 971 (2018); Nano Lett. **16**, 2945-2950 (2016)] (Figs. R5a and R5b). We use a biexponential fit function to examine the decay of population (see Supplementary Note 4 for more details). The fast and slow components of decaying dynamics are denoted as τ_{fast} and τ_{slow} in Fig. R5b, respectively. We choose an excitation energy of 1.75 eV, which is in resonance with X_2 . We investigate the F dependence of τ_{fast} and τ_{slow} for various fillings ν from -2 to 2. Upon a relatively small F of $7.2 \mu\text{J}/\text{cm}^2$, we extract τ_{fast} of 0.44 ± 0.2 ps when $V_G = 0$ V and (Fig. R6a). With a high F of $36.2 \mu\text{J}/\text{cm}^2$, τ_{fast} increases to a value of 0.49 ± 0.04 ps. While a similar F dependence is observed for each filling, the noticeable changes are seen at $V_G = 4$ V, and the F dependence is almost negligible at $V_G = 2$ V. The observed behavior is consistent with the hot phonon effect [Superlatt. Microstruct. **6**, 293-302 (1989); Phys. Rev. B **83** (2011); ACS Nano **8**, 10931-10940 (2014)], where a substantial phonon population is induced by the elevated lattice temperature, which prevent the photoexcited carriers from being cooled via phonon emission. This effect contradicts with the carrier-carrier scattering process, where the sub-picosecond relaxation is known to become faster with increasing F . In prior studies [ACS Nano **8**, 10931-10940 (2014); ACS Nano **4**, 2695-2700 (2010)], A_{1g} and/or E_{2g}^1 optical phonons are proposed as the candidates in the cooling process within a time scale of 0.5 ps. Meanwhile, the modulation of optical phonons frequency and intensity through the carrier doping is reported by previous studies [Nano Lett. **13**, 6170-6175 (2013); Small **13** (2017);

ACS Nano **5**, 5273–5279 (2011)], which may lead to different values of τ_{fast} and the dependency in various ν 's.

The slow decay component τ_{slow} is on the order of ns, which represent an effective lifetime of interlayer excitons in R-stacked WSe₂/WS₂ heterobilayers (Fig. R6b). When $V_G = 0$ V and $F = 7.2$ $\mu\text{J}/\text{cm}^2$, the measured τ_{slow} is a value of 3.38 ± 0.33 ns. This time constant does not match with recent studies [Nat. Mater. **19**, 617-623 (2020); ACS Nano **14**, 4618-4625 (2020)] of reporting relatively short interlayer exciton lifetime of ~ 1 ns for R-stacked WSe₂/WS₂ heterostructures. The discrepancy may arise because our experiments were performed at a temperature of 4 K, minimizing the non-radiative recombination [Nat. Mater. **19**, 617-623 (2020); 2D Mater. **4** (2017)]. With increasing V_G , the extracted τ_{slow} is 2.24 ± 0.19 ns at $V_G = 2$ V and 2.25 ± 0.23 ns $V_G = 4$ V, respectively, which are further decreased compared to the neutral regime. This is consistent with the study [Nat. Phys. **19**, 1286-1292 (2023)] reporting a decrease in the radiative lifetime of interlayer exciton in WSe₂/WS₂ when more n-doping is introduced. Interestingly, no significant F dependence is observed. It suggests a negligible exciton-exciton annihilation process due to the minimal density of photoexcited interlayer exciton. However, in the case of p-doping, τ_{slow} not only strongly depends on F , but also exhibits opposite F dependencies for $V_G = -2.3$ V and $V_G = -3.9$ V. Assuming the radiative decay rate remains relatively unchanged throughout the p-doping [Nat. Phys. **19**, 1286-1292 (2023)], we can infer that the non-radiative decay and the related processes undergo significant changes, particularly due to gap opening at each insulating state, as previously observed in WSe₂/MoS₂ heterobilayers [Nat. Mater. **22**, 605-611 (2023)].

We include the above discussions on the possible origins of τ_{fast} and τ_{slow} in the revised manuscript. The results are discussed in the main text, specifically in Figs. 5a-5c. Further details about the fit function utilized are included in Supplementary Note 4 of the revised manuscript.

Figure R5 | Spectral resolved and integrated $\Delta R/R_0$ signals. **a,b** The measured early- and long-delay $\Delta R/R_0$ spectra (a) and the spectrally integrated $\Delta R/R_0$ (b) as a function of time delay at $V_G = 4$ V, showing a temporal evolution of the carrier population.

Figure R6 | V_G -dependent nonequilibrium decaying dynamics. **a,b** Fast (τ_{fast}) and slow (τ_{slow}) decay time constants as a function of V_G under three different F 's. The black, red and green circle correspond to the different F 's of 7.2, 14.5, and 36.2 $\mu\text{J}/\text{cm}^2$, respectively. Each integer filling ν from -2 to 2 is represented by dashed lines.

We propose that the observed decaying dynamics of τ_{fast} shows the carrier cooling dominated by hot phonon effects. Meanwhile, τ_{slow} shows 4-order longer dynamics than τ_{fast} , which is directly related to the lifetime of interlayer excitons. Distinct F dependence is evident when $V_G < 0$ V, while no significant change is observed when $V_G \geq 0$ V.

Comments 1-5: For the filling factor dependent time-resolved experiments, the pump photon energy is tuned to 2.0 eV (extended Fig.5). Why is the photon energy chosen to be different as compared to Fig. 4 (1.75eV). For the 1.75eV, I can understand that the idea is to only excite WSe₂... this simplifies the possible scattering dynamics of excitons and charge carriers. Why is the photon energy of 2.0eV chosen for the filling dependent experiment? My guess is that the filling factor dependent dynamics is different for different exciton photon energies?

Response 1-5: We appreciate the thoughtful observation made by the Reviewer 1 regarding the choice of pump photon energies in our filling-dependent time-resolved experiments, as depicted in Extended Fig. 5 (Supplementary Fig. 5 in the revised manuscript). When we prepare the original manuscript, we thought that it would be better to employ a pump photon energy of 2.0 eV (in contrast to the 1.75 eV used in Figs. 4a and 4b), simply because it is the same photon energy as the photoluminescence measurement (where we used a 632.8 nm continuous-wave diode laser). We believed that by maintaining the excitation pump energy same as the photoluminescence measurement, readers might be able to treat the observed polarization switching (Fig. 2c) and the enhanced Pauli blocking on an equal footing.

However, after having received the Reviewer 1's comment, we agree that that employing different excitation photon energies could yield different filling factor-dependent dynamics when comparing the resonant and the non-resonant pump excitation. Indeed, a prior study [Nat. Commun. **11**, 5277 (2020)] has highlighted the differences in rise dynamics which vary depending on the excitation energy. In the course of performing additional measurements, we have also realized that these phenomena are evident in our experiments when using different excitation energy. Specifically, when a higher photon energy is used as a pump, we have noticed an increase in the rise time (Fig. R7). This can be attributed to the extended scattering time required for the transition from the continuum to the lowest exciton state.

On the other hand, it is worth of noting that, with sufficiently long time delays, the spectral shift trend appears to be independent of the excitation pump-photon energy. A comparative analysis between Supplementary Fig. 5 in the revised manuscript (2.0 eV excitation energy) and Figure R8 (1.75 eV excitation energy) reveals this similarity in the filling-dependent spectral shifts. We have reproduced the same contour plots of Fig. 4d with lower photoexcitation of 1.75 eV in device R3 (Fig. R9). This yields the consistent signature supporting the dynamic exciton site transition in a moiré supercell. This consistency also implies that, despite the initial differences in

dynamics, the system converges to similar behaviors over long time scales (e.g. $\Delta t = 2$ ns). We hope that this clarification resolves the Reviewer 1's concerns.

Figure R7 | The rise dynamics of population in different pump energies. The normalized temporal evolutions of spectral integrated population dynamics are measured with increasing pump energy. The black, red, and green solid lines correspond to the pump energy of 1.68, 1.72, and 2.00 eV, respectively. The black arrow indicates the longer rise time for higher pump energy.

Figure R8 | Fluence-dependent transient $\Delta R/R_0$ contour plots with various integer fillings. Full fluence-dependent contour plots with integer fillings ν from -2 to 2. The spectrally integrated signal reflects the population of photoexcited carriers. From the temporal evolution of this population, rise and decay time constants are extracted, which are summarized in Fig. 5 of the main text.

Figure R9 | Fluence-dependent transient $\Delta R/R_0$ spectra at time delay of 2 ns. The contour plots measured at $\Delta t = 2$ ns are shown as a function of probe photon energy and V_G to monitor the interlayer exciton site transition. The pump photon energy of 1.75 eV is employed, which is lower than the one used in Fig. 4d. Despite the variation of the pump photon energy, an enhanced Pauli blocking signal is consistently observed in the n-doping regime across all F 's.

Comments 1-6: Starting line 176, the authors state that the early time dynamics is not dependent on the filling factor. I think it would be important and highly interesting to show such data. Why is the charge transfer and ILX formation dynamics not dependent on the filling factor?

Response 1-6: We thank for Reviewer 1's thoughtful comments on the filling-dependent early dynamics. Regarding the sentence, starting line 176 "Akin to Figs. 4a and 4b, no significant spectral re-shaping is observed in the $\Delta R/R_0$ spectra after a sufficient time delay of around 0.1 ns.", We apologize for any confusion caused by the phrasing; our intention was to show no spectral re-shaping for the long time delay, particularly at 2 ns. We would like to clarify that our statement regarding the early time dynamics (not being dependent on the filling factor) is unintentionally misleading. We indeed observe significant spectral re-shaping during the early dynamics, and we recognize that it is influenced by the filling factors, as Reviewer 1 has correctly pointed out. We have already presented the experimental data of charge transfer dynamics in Fig R1. However, measuring the interlayer exciton formation dynamics is challenging due to the small oscillator strength of interlayer excitons. The recent study [Nat. Commun. **14**, 7273 (2023)] exhibits direct measurements of interlayer exciton formation with a much higher resolution and better stable conditions than ours. We attempted follow the reference method during our preparation of revision, but unfortunately, we are unable to provide direct data on the filling-dependent interlayer exciton formation dynamics. We fully agree with Reviewer 1's suggestion that exploring filling-dependent interlayer exciton formation dynamics would be highly important. In Fig. 4d in revised manuscript, however, our focus is on the long time delay (e.g., 2 ns). We demonstrate that interlayer charge transfer dynamics become longer with increasing doping, but the dynamics remain almost unchanged below 1 ps. Following this reason (i.e. at a long pump-probe delay of 2 ns), we believe that the influence of doping on the early dynamics could be negligible, especially when considering the sufficiently long time scale of ns.

Comments 1-7: In how are is the early and long-time dynamics pump power dependent (i.e., dependent on the exciton density)?

Response 1-7: We appreciate Reviewer 1's attention to this aspect, and we would like to confirm that we have addressed this comment in our Response 1-4. There, we have discussed the influence of the pump intensity on the observed dynamics, including the exciton density-related effects. Please refer to the Fig. R6 and Fig. R8 for the corresponding data. We have added the plots to Figs. 5a-5c in the revised manuscript and Supplementary Fig. 6.

Comments 1-8: Are similar data as shown in extended Fig. 5 as well observed if the sample temperature is increased? Is there a critical temperature to observe such dynamics?

Response 1-8: We sincerely appreciate Reviewer 1's insightful inquiry regarding the temperature dependence of the observed dynamics. Following Reviewer 1's advice, we have performed the temperature dependent measurements. Indeed, we find the temperature-dependent behavior to be very interesting. The additional experiments, shown the details in Fig. R10, are performed the temperature range from 40 K to 200 K. Our findings are added in revised Supplementary Fig. 8.

We have found that the spectral shifts, particularly the blueshift at $\nu = 2$ and the redshift at $\nu = -2$, persist at 40 K (Fig. R10a). As the temperature is increased, we observe that these spectral shifts become smaller. For example, the blueshift is almost vanished at 200 K (Fig. R10e). The observed trends and the diminishing effect at higher temperatures, around 200 K, may provide interesting insights into the microscopic origin of the spectral shifts. Based on the fact that the thermal-activation temperature of Mott-Hubbard gap is around 150-180 K in WSe₂/WS₂ [Nature **579**, 353-358 (2020)], the exciton site transition dynamics are also closely related to correlation-driven effects.

Figure R10 | Temperature-dependent transient $\Delta R/R_0$ contour plots with various integer fillings. a-e, $\Delta R/R_0$ contour plots as a function of the probe photon energy and the time delay are illustrated at temperatures 40, 80, 120, 160, and 200 K, respectively. An enhanced blueshift signal at $\nu = 2$ is evident at 40 K, particularly after the long time delay (e.g. 2 ns). This finding is attributed to the correlation-driven exciton site transition, a topic elaborated in the main text. With increasing the temperature up to 200 K, the distinct blueshift signature gradually diminishes. When the temperature reaches 200 K, the ν -dependent spectral shift exhibits marginal changes. Based on the fact that the thermal-activation temperature for Mott-Hubbard gap is around 150-180 K in WSe_2/WS_2 [Nature **579**, 353-358 (2020)], it suggests that the microscopic origin of the dynamics of exciton site transition is related to correlation effects.

Comments 1-9: Do I understand correctly that the time-dependent blue-/redshift is a consequence of the Coulomb repulsion of the doped electrons and the photoexcited ILX? If so, can this be made more clear by labelling spectral features or trends in the figures? In how are does the time-scale of the blue-/redshift agree with this explanation? Hence, in how are is it a non-equilibrium response?

Response 1-9: We are grateful for Reviewer 1's important questions. Regarding the Comments about the time-dependent blue-/redshift, we acknowledge the importance of providing a clear indication of spectral features. Following Reviewer's suggestions, we have added a schematic inset in the revised manuscript Fig. 4d (See Fig. R11). This inset illustrates the blueshift of the absorption spectrum, helping to visually highlight the trend caused by Pauli blocking. This trend, which is clearly observed at the long delay where the dynamics of the interlayer exciton are exclusively seen, is a consequence of the site transition, resulting from the Coulomb repulsion of the doped electrons and the photoexcited interlayer exciton, as mentioned in the Comments. While the clear spectral shift is observed at the long delay, we have addressed the early nonequilibrium dynamics of the site transition. Specifically, in the revised manuscript, we have included discussions (page 12-14) and figures (Fig. 5) related to the site transition time, particularly emphasizing its asymmetric dependence on doping. This analysis implies that the site transition occurs exclusively upon n-doping case. We believe that the related concerns have been addressed in both Response 1-3 and Response 1-4. We hope these modifications provide a clearer and more detailed understanding of the spectral features and their relation to the Coulomb repulsion.

Figure R11 | Revised version of Fig. 4d. An inset has been added to provide clarification on the spectral blueshift signal caused by Pauli blocking.

General remarks and comments of Reviewer 2:

The manuscript by Kim et al reports a study on the polarization resolved optical spectra of twisted WSe₂/WS₂ hetero-stacking. The authors observed polarization switching and absorption edge shifts in near-zero angle aligned WSe₂/WS₂ heterobilayers. They attributed the observation to correlation-driven effects. Overall, the authors presented certain new findings (by both of CW and ultrafast laser excited methods), and I appreciate the experimental data quality; but in my opinion it is too “technical” to falls into a specified area, which may not attracts broad interesting in the community. Although it is well organized and clearly written, the manuscript calls for in-depth analysis or insightful physics for publishing in Nature Communications.

Response 2: We appreciate Reviewer 2’s time, consideration, and careful comments. We thank Reviewer 2 for the positive assessment of the quality of our work. We agree with his/her suggestion that more in-depth analysis or insightful physics is necessary to enhance the manuscript's suitability for publication in Nature Communications. We carefully consider his/her comments, and have performed an extensive revision, such as fabricating an additional twisted TMD device, measuring the early pump-probe dynamics, and provide more quantitative analysis.

Below we present our point-by-point responses to the comments raised by Reviewer 2.

Comments 2-1: My major concern is about the working mechanism. The authors stated that the polarization switching results from strong correlation induced exciton site transition. It is not clear to me what is the role played by the correlation effect. How does it affect the site transition in real space? And why does it occur at/around the “Mott insulator” state? It would be necessary to clearly point out the physics picture behind.

Response 2-1: We appreciate Reviewer 2’s thoughtful and insightful comments regarding the working mechanism of the polarization switching and the correlation-induced exciton site transition in our study. It has prompted us to provide a clearer and more comprehensive explanation of the role played by correlation effects in the exciton site transition.

In our study, we refer to the correlation effect as the on-site Coulomb repulsion U . This parameter becomes particularly crucial in the moiré superlattice with a large lattice constant of approximately 8 nm in our case. Due to this large lattice constant, the electron hopping (kinetic energy) is significantly smaller than U . When electrons occupy the moiré unit cell in a one-to-one correspondence ($\nu = 1$) by electrostatic doping, the emergence of Mott insulating state is triggered by on-site Coulomb repulsion (correlation). The intriguing physics manifests when additional electrons are doped into the lattices, i.e., for $\nu \geq 1$. There are two scenarios. First, the extra electron may occupy the same orbital as the former one with an opposite spin, resulting in both electrons occupying the same site. Second, when U exceeds the energy differences between two moiré local minimums (ΔE_g), the extra electron avoids double-occupancy and resides at the second local minimum site. In this case, the correlation lead to a charge transfer insulating state. This is the reason why the site transition and polarization switching occur at/around $\nu = 1$. Although there has been debate whether WSe₂/WS₂ is a Mott insulator or a charge transfer insulator, recent experimental and theoretical studies [Nat. Phys. **19**, 1286-1292 (2023); Phys. Rev. B **102** (2020)] support to the latter.

It is important to note that “the Coulomb repulsion” has a different meaning when referred to the charge transfer insulator and our case. Specifically, the electron-electron repulsion U_{e-e} is used for the charge transfer insulators, whereas the electron-exciton repulsion U_{e-ex} is applicable in our case. Despite this different meaning, we can treat them similarly due to the type II band alignment, where electrons and holes are separated in different layers. Because U_{e-ex} is dominated by a short-range Coulomb interaction, interactions within the same layer are significantly stronger than those involving particles from different layers. Consequently, we expect that the energy scale of U_{e-ex} and U_{e-e} is on the same order [Science **380**, 860-864 (2023)]. The site transition of interlayer

excitons is indicated by polarization switching (please see Supplementary Table 1-2 for detailed information on site-dependent polarization). We hope this clarification provides a more explicit connection between the correlation effect, exciton site transition, and the observed polarization switching phenomenon. In the Supplementary Information, Supplementary Note 2 provides quantitatively estimated relationship between ΔE_g and U_{e-ex} .

Comments 2-2: Another flaw of this manuscript is about the filling factor calculation. Generally, filling factors for a Moire hetero-stacking are calculated from the area of Moire super cell. It looks that the authors assigned the filling factors according to the optical spectrum. It is quite strange that the filling factors are not equally spaced. The assignment is somehow arbitrary. And it hinders the validity of the conclusion.

Response 2-2: First of all, we apologize that we have not provided details of how the filling factors are determined. From a theoretical standpoint of view, the filling factors are determined by calculating the moiré supercell density. Please note that following discussion is in the method of revised manuscript. The moiré density n_0 is given by $n_0 = 2/(\sqrt{3}a_M^2)$ for a triangular superlattice and the filling factor ν is given by $\nu = n/n_0$. Here a_M is the moiré superlattice constant which is determined by the twist angle (θ) and lattice mismatch $\delta = (a_{Se} - a_S)/a_{Se}$ between WSe₂ and WS₂, where a_{Se} of 0.328 nm is the lattice constant of WSe₂ and a_S of 0.315 nm is the one of WS₂. We obtain $a_M \sim 7.54$ nm by using the relation $a_M = a_{Se}/\sqrt{\delta^2 + \theta^2}$. Corresponding moiré density n_0 is 2.19×10^{12} cm⁻² for $\theta = 0.98^\circ$ (Fig. R12). This heterostructure is enclosed by the top and bottom hexagonal boron nitride (hBN) dielectric layers, whose thickness are 17 nm and 25 nm confirmed by atomic force microscopy (Fig. R13). The electron density (n) is obtained from the parallel-plate capacitor model as $n = \epsilon\epsilon_0 V_G/d_{\text{bottom}}$, where ϵ is dielectric constant of hBN and d_{bottom} is the thickness of bottom hBN. Based on this relation, we estimate $\nu = 1$ when $V_G = 2.5 \pm 0.5$ V. The relatively large error is attributed to uncertainties in ϵ , d_{bottom} , and θ . From an experimental view, one approach involves measuring the abrupt changes of resistance to examine the Mott or band insulating state, as shown in a previous study [Nature 579, 353–358, 2020]. In the insulating states, the reflection contrast (RC) is measured to be large at the photon-energy peak of moiré excitons because of the quenched free-carrier screening of electron-hole interactions. This results in the partial restoration of the oscillator strength and the reflection contrast for the neutral exciton resonance. To quantify this approach, we measure the peak amplitude of the reflection contrast (RC) as a function of V_G . The results are shown in Fig R14. In this approach, we estimate $\nu = 1$ at $V_G = 2$ V and identify several insulating states by inspecting the V_G dependent peaks of the RC spectra. Indeed, we follow the above experimental approach and these estimations are consistent with the theoretical calculations with considering the uncertainties. The results are described in method and Supplementary Fig. 1 in the revised manuscript.

Figure R12 | Determination of crystal orientation by angle-resolved second harmonic generation spectroscopy. Solid red circles, green triangles, and black rectangles are the polarization-dependent SHG data from monolayer WSe₂, WS₂, and WSe₂/WS₂ heterobilayer regions, respectively. The solid curve is a fit to the data using $y = y_0 + A \sin^2 \left[\frac{\pi}{w} (x - x_c) \right]$, where x is the excitation polarization angle and x_c is the crystal orientation. While w is fixed with 60°, y_0 and A are the free fitting parameters. The twist angle between WSe₂ and WS₂ is determined to be $0.98^\circ \pm 0.28^\circ$ for the device R3. When the two monolayers are R-stacked (nearly 0°), SHG from the heterobilayer is much stronger than the monolayer.

Figure R13 | Atomic force microscopy measurement for bottom and top hexagonal boron nitride. a, b Line-profile along the interface between hBN and the substrate. The red arrows show that the estimated thicknesses of the bottom (**a**) and top (**b**) layer are 25 nm and 17 nm, respectively

Figure R14 | Determination of filling factors. **a** V_G -dependent reflection contrast spectrum. The peak of absorption shows the kinks and the splitting at the integer ν 's. **b** Reflection contrast peak amplitude at the lowest-energy exciton resonance as a function of V_G . The strong peak amplitude is observed for each integer ν (dashed black lines) and they are equally spaced with a value of 2 and 1.6 V for n- and p-doping, respectively.

Comments 2-3: And in addition, a minor point: labels of d-f are missing in figure 2.

Response 2-3: We sincerely apologize any confusion caused by the absence of labels of d-f in Fig. 2. We have promptly corrected this mistake and revised the manuscript to avoid similar errors in the future.

REVIEWER COMMENTS

Reviewer #1 (Remarks to the Author):

I would like to thank the authors very much for their reply to my comments, and suggest that the manuscript can be published in Nature Communications as is.

Reviewer #2 (Remarks to the Author):

This is the second-round review of the manuscript "Correlation-driven nonequilibrium exciton site transition in a WSe₂/WS₂ moiré supercell". I appreciate the authors' effort as well as their willingness to improve the manuscript.

However, I am not fully convinced by the authors' reply.

Firstly, about the working mechanism, the authors did not revise the manuscript although they gave a clear explanation. I would suggest the author include a brief discussion in the maintext.

And secondly, about the filling factor. I am still confused why the filling factor is different for electron and hole. According to the parallel-plate capacitor model, they should be same. And also, what is the zero-doping level for the device. There are two lines for zero filling. The authors should clearly discuss the uncertainty if it results from electron-hole puddles or trapping states.

Point-by-point responses to the issues raised by the reviewers

General remarks and comments of Reviewer 1:

I would like to thank the authors very much for their reply to my comments, and suggest that the manuscript can be published in Nature Communications as is.

Response 1:

We thank Reviewer 1's time to review our manuscript. We sincerely appreciate his/her positive comment and suggestion regarding the publication of our work. Reviewer 1's support is truly encouraging, and we are grateful for valuable input throughout the review process.

General remarks and comments of Reviewer 2:

This is the second-round review of the manuscript "Correlation-driven nonequilibrium exciton site transition in a WSe₂/WS₂ moiré supercell". I appreciate the authors' effort as well as their willingness to improve the manuscript.

However, I am not fully convinced by the authors' reply.

Response 2: We appreciate Reviewer 2's thoughtful comments, continued engagement with our manuscript, and acknowledgment of our efforts. We acknowledge that there may still be some aspects of the manuscript requiring clarification or improvement. We sincerely apologize if our response did not fully address Reviewer 2's concerns. Considering his/her comments, we have revised the manuscript to include clear discussions and have conducted additional analyses to provide more detailed explanation regarding the filling factors.

Below we present our point-by-point responses to the comments raised by Reviewer 2.

Comments 2-1: Firstly, about the working mechanism, the authors did not revise the manuscript although they gave a clear explanation. I would suggest the author include a brief discussion in the maintext.

Response 2-1: We appreciate Reviewer 2's thoughtful suggestions for improvement. In response to comments regarding the working mechanism, we have revised the manuscript and have included a brief discussion in the main text to provide further clarity. In the revised manuscript, we have corrected the paragraph (page 7, line 133-145) as below. We hope that this revision addresses his/her concerns satisfactorily.

Before: Interestingly, the threshold voltage is close to the Mott insulating state ($\nu = 1$). This implies that different correlation effects are set in, compared to the p-doping case. The polarization switching (from the co-circularly polarized to the linearly-polarized PL) cannot be understood by the valley-flip scattering; it does not add any additional polarization switching mechanisms in the n-doping case¹⁸. Rather, we put the strong correlation-induced moiré exciton site transition as the key contributor, which is in line with the fact that the WSe₂/WS₂ heterobilayer is indeed a charge transfer insulator^{11,21,22} for $\nu > 1$.

After: (page 7, line 133-145) Interestingly, the threshold voltage is close to the Mott insulating state ($\nu = 1$) where electrons occupy the moiré unit cell in a one-to-one correspondence triggered by on-site Coulomb repulsion (U_{e-e}). The polarization switching (from the co-circularly polarized to the linearly-polarized PL) cannot be understood by the valley-flip scattering; it does not add any additional polarization switching mechanisms in the n-doping case¹⁸. Rather, we consider two scenarios when additional electrons are doped into the lattices, i.e., for $\nu \geq 1$. First, the extra electron may occupy the same orbital as the former one with an opposite spin, resulting in both electrons occupying the same site. Second, when U_{e-e} exceeds the energy differences between two moiré local minimums (ΔE_g), the extra electron avoids double-occupancy and resides at the second local minimum site. In this case, the correlation lead to a charge transfer insulating state. We put the strong correlation-induced moiré exciton site transition as the key contributor for polarization switching, which is in line with the fact that the WSe₂/WS₂ heterobilayer is indeed a charge transfer insulator^{11,21,22} for $\nu > 1$.

Comments 2-2: And secondly, about the filling factor. I am still confused why the filling factor is different for electron and hole. According to the parallel-plate capacitor model, they should be same. And also, what is the zero-doping level for the device. There are two lines for zero filling. The authors should clearly discuss the uncertainty if it results from electron-hole puddles or trapping states.

Response 2-2: We sincerely appreciate Reviewer 2's insightful comments regarding the filling factor asymmetry observed between electron- and hole-doping regime. As highlighted by Reviewer 2, the filling factor gap, meaning the voltage difference between each integer filling factor, deviates for electrons and holes across all our devices despite the expected same gap according to the parallel-plate capacitor model. For a comprehensive overview, please refer to Table R1, which summarizes the filling gaps for our R-stacked devices, while detailed data on filling factor determinations are presented in Figs. R1, R2, and R3. We fully agree with the raised concern regarding this asymmetric feature and its potential implications. Importantly, please note that this electron-hole asymmetry has also been reported by many prior studies in transition metal dichalcogenides (TMDs) heterobilayers [Nat. Nanotechnol. **16**, 52-57 (2021); Nat. Mat. **22**, 599-604 (2023); Nat. Commun. **12**, 3608 (2021); Nat. Nanotechnol. **18**, 233-237 (2023); Nature **579**, 353-358 (2020); Nat. Phys. **20**, 34-39 (2024); Phys. Rev. Lett. **127**, 037402 (2021)]. Below we present the origins for this phenomenon.

While various factors may contribute to this result, we suggest that the primary factor is related to the contact issues. Establishing contact between the semiconductor and metal (or semi-metal) has long been a challenge in conducting transport measurements at cryogenic temperatures. This is because it leads to a Schottky barrier, significantly reducing the transport efficiency of carriers. Namely, a higher Schottky barrier requires a larger gate voltage to overcome the barrier for the same current flow. This effect is significantly influenced by whether electron or hole doping is introduced, depending on the work function of the contact materials. For instance, platinum, with one of the highest work function materials, forms excellent ohmic contact for hole doping, but the high Schottky barrier is formed in electron doping. Consequently, while many fascinating quantum transport experiments in TMDs are exclusively conducted with hole doping using platinum contact [Nature **597**, 350-354 (2021); Nature **600**, 641-646 (2021); Nature **622**, 74-79 (2023); Nature **616**, 61-65 (2023); Nature **622**, 69-73 (2023); Nat. Nanotechnol. **19**, 28-33 (2024); Nat. Phys. **20**, 275-280 (2024); Phys. Rev. X **14**, 011004 (2024)], to the best of our knowledge, equivalent measurements in electron doping have not been reported due to challenges in establishing good

contact. In our case, we have utilized few layers of graphene contact, which also exhibits electron-hole asymmetry issues. On the other hand, establishing a good contact in twisted bilayer graphene devices is relatively easier compared to semiconductors TMDs. As a result, it is well established that the filling gaps in graphene-based samples are evenly spaced, leading to the emergence of numerous intriguing quantum phenomena in both electron and hole doping regimes [Nature **556**, 80-84 (2018); Nature **556**, 43-50 (2018); Nature **573**, 91-95 (2019); Nature **574**, 653-657 (2019); Nature **588**, 610-615 (2020); Nature **589**, 536-341 (2021); Nat. Phys. **16**, 520-525 (2020); Science **363**, 1059-1064 (2019)]. Even though electron-hole puddles have been observed in graphene [Nat. Phys. **4**, 144-148 (2008)], evenly spaced filling gaps are still observed in graphene-based samples. This observation suggests that the effect of puddles on asymmetric filling gaps can be excluded. However, trapping states and various defect states in hBN [Nat. Nanotechnol. **11**, 37-41 (2016); Nat. Commun. **13**, 3233 (2022); Phys. Rev. B **97**, 064101 (2018); Phys. Rev. B **102**, 099903 (2020)] or TMDs [Nat. Commun. **10**, 3825 (2019); Sci. Adv. **3**, e1701661 (2017); Nat. Commun. **13**, 492 (2022)] may contribute to the electron-hole asymmetry with the contact issues. Furthermore, beyond the electron-hole asymmetry, other studies have reported distinct filling gaps in the single-carrier doping regime. For instance, differences in the filling gap between the transition from $\nu = 0$ to $\nu = 1$ and $\nu = 1$ to $\nu = 2$ have been reported [Nat. Mat. **22**, 599-604 (2023); Nat. Phys. **20**, 34-39 (2024)]. This phenomenon may be attributed to multi-orbital effects. Particularly, for $\nu > 1$, prior study indicates that carriers can occupy not only the correlated moiré mini bands, but also other orbitals that are unrelated to the correlated physics. [Phys. Rev. B **99**, 125424 (2019)].

Next, we turn the discussion to zero fillings lines. We plot two lines for zero fillings, representing the neutral regime of the device (Figs. R1, R2, and R3). The presence of two lines for zero filling arises from defining the gate voltage range where the Fermi level lies within the energy gap of the type-II band alignment. Specifically, the upper (lower) zero line indicates the Fermi level touching the edge of the conduction (valence) band. These double lines have also been consistently observed in prior studies [Nature **579**, 353-358 (2020); Nature **609**, 52-57 (2022)], providing a consistent explanation.

Device	Electron filling gap (V)	Hole filling gap (V)
R1	1.6	1.9
R2	1.3	1
R3	2	1.6

Table R1 | The filling gaps in electron and hole doping are summarized for devices R1, R2, and R3. These devices share the same stacking order, with WSe₂ positioned above the WS₂ layer.

Figure R1 | Determination of filling factors on device R3. **a** V_G -dependent reflection contrast spectrum. The peak of absorption shows the kinks and the splitting at the integer ν 's. **b** Reflection contrast peak amplitude at the lowest-energy exciton resonance as a function of V_G . The strong peak amplitude is observed for each integer ν (dashed black lines) and they are equally spaced with a value of 2 and 1.6 V for electron- and hole-doping, respectively.

Figure R2 | Determination of filling factors on device R1. a,b Same plots as Fig. R1, but for the device R1. The filling gaps are equally spaced with a value of 1.6 and 1.9 V for electron- and hole-doping, respectively.

Figure R3 | Determination of filling factors on device R2. a,b Same plots as Fig. R1, but for the device R2. The filling gaps are equally spaced with a value of 1.3 and 1 V for electron- and hole-doping, respectively.

REVIEWERS' COMMENTS

Reviewer #2 (Remarks to the Author):

The authors properly addressed all issues. I have no further question.

Point-by-point responses to the issues raised by the reviewers

General remarks and comments of Reviewer 2:

The authors properly addressed all issues. I have no further question.

Response 2: We thank for Reviewer 2's positive feedback and confirmation that all issues have been properly addressed.